# MS CD49d^+^CD154^+^ Lymphocytes Reprogram Oligodendrocytes into Immune Reactive Cells Affecting CNS Regeneration

**DOI:** 10.3390/cells8121508

**Published:** 2019-11-25

**Authors:** Paweł Piatek, Magdalena Namiecinska, Małgorzata Domowicz, Patrycja Przygodzka, Marek Wieczorek, Sylwia Michlewska, Natalia Lewkowicz, Maciej Tarkowski, Przemysław Lewkowicz

**Affiliations:** 1Department of Neurology, Laboratory of Neuroimmunology, Medical University of Lodz, Poland, Pomorska Str. 251, 92-213 Lodz, Poland; pawel.piatek@umed.lodz.pl (P.P.); magdalena.namiecinska@umed.lodz.pl (M.N.); malgorzata.domowicz@umed.lodz.pl (M.D.); 2Institute of Medical Biology, Polish Academy of Sciences, 93-232 Lodz, Poland; przypat@gmail.com; 3Department of Neurobiology, Faculty of Biology and Environmental Protection, University of Lodz, 90-236 Lodz, Poland; marek@biol.uni.lodz.pl; 4Laboratory of Microscopic Imaging and Specialized Biological Techniques, Faculty of Biology and Environmental Protection, University of Lodz, 90-236 Lodz, Poland; sylwia.michlewska@biol.uni.lodz.pl; 5Department of General Dentistry, Medical University of Lodz, 92-213 Lodz, Poland; natalia.lewkowicz@umed.lodz.pl; 6Department of Biomedical and Clinical Sciences, Luigi Sacco, University Hospital, University of Milan, 20122 Milano, Italy; maciej.tarkowski@outlook.it

**Keywords:** oligodendrocyte precursor cells, remyelination, multiple sclerosis, myelin-specific lymphocytes

## Abstract

The critical aspect in multiple sclerosis (MS) progression involves insufficient regeneration of CNS resulting from deficient myelin synthesis by newly generated oligodendrocytes (OLs). Although many studies have focused on the role of autoreactive lymphocytes in the inflammatory-induced axonal loss, the problem of insufficient remyelination and disease progression is still unsolved. To determine the effect of myelin-specific lymphocytes on OL function in MS patients and in a mouse model of MS, we cultured myelin induced MS CD49d^+^CD154^+^ circulating lymphocytes as well as Experimental Autoimmune Encephalomyelitis (EAE) mouse brain-derived T and memory B cells with maturing oligodendrocyte precursor cells (OPCs). We found that myelin-specific CD49d^+^CD154^+^ lymphocytes affected OPC maturation toward formation of immune reactive OLs. Newly generated OLs were characterized by imbalanced myelin basic protein (MBP) and proteolipid protein (PLP) production as well as proinflammatory chemokine/cytokine synthesis. The analysis of cellular pathways responsible for OL reprogramming revealed that CD49d^+^CD154^+^ lymphocytes affected miRNA synthesis by dysregulation of polymerase II activity. miR-665 and ELL3 turned out to be the main targets of MS myelin-specific lymphocytes. Neutralization of high intracellular miR-665 concentration restored miRNA and MBP/PLP synthesis. Together, these data point to new targets for therapeutic intervention promoting CNS remyelination.

## 1. Introduction

Dysfunction of oligodendrocytes (OLs), the cells responsible for neuron myelin sheath formation, is suggested as one of the most important factors underlying the incomplete remyelination in multiple sclerosis (MS) pathology. In the course of relapsing-remitting MS (RR-MS) there are observed acute attacks (relapses) followed by a period of partial withdrawal of symptoms (remission) [1]. Relapse mainly involves the brain areas previously affected by the disease that leads to gradual accumulation of irreversible impairment due to extended local demyelination [2]. The crucial cause of slowly progressing, irreversible disease-related disability is a renewal of only fragmentary tissue [3].

Although some studies highlighted the participation of oligodendrocytes (OLs) as the principle cells responsible for neuron myelin sheath formation, their critical role in incomplete remyelination in MS pathology is still not clarified [3,4]. In adults, oligodendroglia precursor cells (OPCs) constitute approximately 6% of the CNS total cell number and are abundant throughout the gray matter of CNS, where they generate new OLs during remyelination [5,6]. The structure of new myelin sheaths formed in remyelinating lesions is thinner than during development [7]. Disproportions in myelin proteins, myelin basic protein (MBP), myelin oligodendrocyte glycoprotein (MOG), and proteolipid protein (PLP) can explain the observed phenomenon. Recent studies demonstrated that approximately 70% of lesions contained progenitors or premyelinating OLs characterized by inability to differentiate into mature cells [8,9]. Thus, the deficiency in myelin synthesis by OLs is thought to be mediated at the early stages of OPC differentiation, probably during migration to the inflammatory site. For its effectiveness, two consecutive processes must occur within a strictly defined period. First, OPCs should be recruited to the demyelinating lesion and then growth factors and cytokines should create a specific environment, which is necessary for directed progenitor differentiation.

The presence of inflammatory cells was revealed in MS lesions regardless of the clinical diagnosis [2]. Recently, we identified a new subpopulation of myelin-specific CD49d^+^CD154^+^ lymphocytes in the peripheral blood of RR-MS patients during remission that proliferated in vitro in response to myelin peptides. These lymphocytes possessed the unique ability to migrate towards maturing oligodendrocyte precursor cells (OPCs) and synthetize proinflammatory chemokines/cytokines [10]. In this study, we explored the role of MS myelin-specific CD49d^+^CD154^+^ lymphocytes in dysregulating remyelination by affecting the transcriptional and post-transcriptional processing of OPC maturation. We demonstrated that myelin-specific lymphocytes induced OPC reprogramming for immune-reactive OLs with impaired myelin synthesis which fueled ongoing accumulation and activation of lymphocytes. This positive feedback reaction loop is critical in creating remyelinating plaques to become more susceptible to reattack contrary to unaffected brain tissue, resulting in deterioration of existing lesions and increased severity of MS symptoms within each new relapse.

## 2. Materials and Methods

### 2.1. Human Subjects

All participants of the study were diagnosed and recruited at the Department of Neurology, Medical University of Lodz. Ten patients (six females, four males; aged 43.2) diagnosed with RR-MS according to the McDonald criteria 2010 [11] were enrolled into the study during disease remission. Mean disease duration was 5.4 years and mean time from last relapse was eight months. Relapse was defined as the appearance of new neurological signs or worsening of pre-existing ones after a minimum 30 days of clinical stability. Remission was defined as minimum three months of neurological stabilization after last relapse. None of the patients received systemic steroids or other anti-inflammatory or immunosuppressive drugs for at least three months prior to the study. For the control group, ten healthy volunteers matched by age and sex were recruited. A total of 20 mL of peripheral blood was collected from the patients during remission phase of the disease and healthy controls (HC). The study was approved by the Ethics Committee of the Medical University of Lodz (RNN/44/14/KE and RNN/28/18/KE), and informed consent was obtained from each participant of the study.

### 2.2. Experimental Autoimmune Encephalomyelitis (EAE) Mouse Model

For the mouse model of MS, C57Bl/6 mice were housed and maintained in an accredited facility, the Animal Core Department of the Medical University of Lodz. Six to eight-week-old female C57Bl/6 mice were injected with MOG_35-55_ (NH2-MEVGWYRSPFSRVVHLYRNGK-COOH) in complete Freund adjuvant (CFA) subcutaneously [12,13,14]. On day 0, each mouse received 0.25 mL of 0.15 mg mixture of dissolved MOG_35-55_ in 0.1 mL of PBS and 0.75 mg of *Mycobacterium tuberculosis* in 0.15 mL of CFA, injected in four abdominal sites. The blood–brain barrier (BBB) was damaged by using 0.2 µg Pertussis toxin (Sigma-Aldrich, St. Louis, MO, USA) injected into a tail vein on days 0 and 2. Mice were observed for neurologic signs of EAE and were scored using the scale 0–5 as follows: 0—nondisease; 1—weak tail or wobbly walk; 2—hind limb paralysis; 3—forelimbs paralysis; 4—hind and forelimb paralysis; 5—death or euthanasia for ethical reasons. On day 14–15 (peak of the disease) or three weeks after the peak (remyelination period), mice were sacrificed and sampled to further analysis. All experiments were approved by the Medical University Ethics Committee (12/ŁB702/2014).

### 2.3. Human Cell Model of Progenitor Cell Differentiation to Mature Myelin-Producing Oligodendrocytes

A human oligodendroglia cell line MO3.13 (OLs, Tebu-bio, Le Perray En Yvelines, Rambouillet, France) was used as the model of progenitor cell differentiation to mature myelin-producing OLs. We chose the MO3.13 cell line, which differentiates after phorbol 12-myristate 13-acetate (PMA) stimulation, as the most adequate model of OPC maturation. Contrary to other OL lines (HOG or KG-1C), during maturation, MO3.13 cells exhibited the strongest similarity to primary human OLs in morphology as well as in gene and protein expression [15]. OLs were cultured in DMEM high glucose medium supplemented with 10% fetal bovine serum, 1% penicillin–streptomycin and maintained at 37 °C with 5% CO_2_ in humidified atmosphere. The cultures were conducted in 75 cm^2^ flasks (Nunc, Thermo Scientific, Waltham, MA, USA). After reaching 80% confluence, cells were passaged by using a 0.25% trypsin–EDTA solution in total three times for a week with a dilution factor of 1/8. To induce OPC maturation, Phorbol 12-Myristate 13-Acetate was added (0.1 mM) for 72 h and cells were incubated at 37 °C with 5% CO_2_ in humidified atmosphere. All reagents used for culturing were purchased from Sigma-Aldrich.

### 2.4. MO3.13 Transfection

A total of 6 × 10^5^ MO3.13 cells were seeded on six-well culture plates (37 °C, 5% CO_2_) one day before transfection to achieve suitable confluence. For transfection, X-treme GENE 9 Reagent (Roche) was used. Cells were transfected with 50 nM miRCURY LNA inhibitors: hsa-miR-21-3p, hsa-miR-665, hsa-miR-21-3p with hsa-miR-665, and Negative Control A which does not recognize any human gene (Exiqon, Vedbæk, Denmark). After 48 h incubation (37 °C, 5% CO_2_), RR-MS myelin-specific CD49d^+^CD154^+^ lymphocytes or HC lymphocytes were added to transfected MO3.13 cells stimulated with PMA (100 nM). After 72 h incubation (37 °C, 5% CO_2_), cells were separated using CD45 MicroBeads (Miltenyi Biotec, Bergisch Gladbach, Germany).

### 2.5. RR-MS CD49d^+^CD154^+^ Lymphocyte Sorting

Peripheral blood mononuclear cells (PBMCs) were isolated by density centrifugation over Lymphoprep (Axis-Shield, Oslo, Norway) according to manual instruction. PBMCs were stimulated by adding mixture of peptides: proteolipid protein (PLP139-151 NH2-HSLGKWLGHPDKF-COOH; pepPLP), myelin oligodendrocyte glycoprotein (MOG35-55 (NH2-MEVGWYRSPFSRVVHLYRNGK-COOH; pepMOG) and myelin basic protein (MBP85-99 NH2-VHFFKNIVTPRTPPP-COOH; pepMBP) (each 25 μg/mL) for 21 h (37°C, 5% CO2 in humidified atmosphere). Following the incubation, cells were washed and labelled with 5 μg/mL CD154-PE (clone 89–76, BD Biosciences, San Jose, CA, USA) and CD49d-FITC (clone 9F10, BD Biosciences, San Jose, CA, USA) mAbs for 30 min at 4 °C. CD49d^+^CD154^+^ population was isolated by FACS sorting procedure using a FACSAria with purity mask mode (BD Biosciences, San Jose, CA, USA). The purity of FACS-sorted CD49d^+^CD154^+^ cell fraction assessed by two-color flow cytometry was ~99.5%.

### 2.6. hOPC–Lymphocyte Coculture

To examine the effect of myelin-specific CD49d^+^CD154^+^ lymphocytes on OPC differentiation, 4 × 10^4^ lymphocytes were added to 2 × 10^6^ OPCs (1:50) which were seeded on six-well plates 24 h earlier. PMA (phorbol 12-myristate 13-acetate), an artificial stimulator of OPC differentiation, was added (0.1 mM) to culture in DMEM high glucose medium supplemented with 5% FBS, 1% penicillin–streptomycin at 37 °C with 5% CO_2_ for 72 h. After incubation, supernatants were collected and cells were washed in DPBS. For further OL analysis, lymphocytes were removed by antihuman CD45 MicroBeads (Miltenyi Biotec, Bergisch Gladbach, Germany). Lymphocytes were attached to the magnetically polarized side of the Eppendorf tube and OLs remained at the tube’s bottom. These procedures allowed us to avoid OL activation during flowing through the magnetic column. OLs were transferred into a new tube, washed, and the dry pellet was frozen in −80 °C for NGS mRNA and miRNA ddPCR analysis.

### 2.7. Isolation of EAE Mouse Memory CD19^+^ and CD3^+^ Brain-Infiltrating Lymphocytes

Mice were perfused intracardially with PBS before the brain dissection and homogenization. Brain mononuclear cells (BMCs) were isolated using 37–70% (*v*/*v*) Percoll gradients. Three weeks after the peak of the disease, EAE BMCs were separated by magnetic microbeads in a two-step procedure: positive selection of memory CD19^+^ cells (mCD19^+^) (Memory B Cell Isolation Kit, Miltenyi Biotec, Bergisch Gladbach, Germany) followed by negative selection of CD3^+^ cells from the remaining unlabeled cells (Pan T cell isolation kit II, Miltenyi Biotec, Bergisch Gladbach, Germany). Four brain tissues were pooled for each experiment. BMC purity and phenotype were analyzed by LSRII flow cytometry (BD) using conjugated anti-B220 (clone RA3-6B2, BioLegend, San Diego, CA, USA), TCR-β (clone H57-597, BioLegend, San Diego, CA, USA), CD3 (clone 17A2, BioLegend, San Diego, CA, USA), CD19 (clone 1D3, Abcam, Cambridge, UK), and appropriate isotype control monoclonal antibodies (mAbs).

### 2.8. Neonatal Mouse Oligodendrocyte Precursor Cell (OPC) Isolation

OPCs were prepared from one-day-old neonatal mouse pup brains. Mice were anaesthetized and brains were removed from the skull cavity. Subsequently, the cortex was rinsed with HBSS, cut and digested by trypsin (0.25%) and DNase I (0.005%) for 15 min (shaking every 5 min). After that, the blocking buffer was added to the tissue and centrifuged at 100× *g* for 2 min. The samples were washed, suspended in DMEM/F12 (supplemented with 10% FBS and streptomycin–penicillin) and cultured for 10 days (37 °C, 5% CO_2_, humid atmosphere). Next, the flasks were shaken at 90 rpm for 40 min to remove microglia, and incubated for next four days with DMEM/F12 medium (supplemented with 3% FBS, 10 μg/mL biotin, 5 μg/mL insulin, N2 and penicillin/streptomycin). All reagents used for culture were purchased from Sigma-Aldrich. The medium was changed every three days following 2–3 weeks. The isolation correctness, purity, and cell differentiation were estimated by O4 (clone O4, R&D Systems, Minneapolis, MN, USA), Nestin (clone Poly18419, BioLegend, San Diego, CA, USA), MOG (clone D10, Santa Cruz Biotechnology, Dallas, TX, USA), and GFAP (clone H50, Santa Cruz Biotechnology, Dallas, TX, USA), expression by immunocytochemical (ICC) analysis, as well as morphology shape in differential contrast microscopy (DIC) examination.

### 2.9. Coculture of Mouse OPCs with EAE CD3^+^ and mCD19^+^ Cells

To examine the effect of EAE CD3^+^ and mCD19^+^ BMCs on mOPC maturation, 4 × 10^4^ lymphocytes, seeded on six-well plates 24 h earlier, were added to 2 × 10^6^ OPCs (1:50). Cells were cultured in DMEM high glucose medium containing: PMA (0.1 mM), 5% fetal bovine serum, and 1% penicillin–streptomycin at 37 °C with 5% CO_2_ in humidified atmosphere for 7 2 h (all reagents purchased from Sigma-Aldrich, St. Louis, MO, USA). After incubation, supernatants were collected and cell suspensions were washed in DPBS. Subsequently, lymphocytes were removed by magnetic antimouse CD45 MicroBead (Miltenyi Biotec, Bergisch Gladbach, Germany), as described in the experiment with human cells.

### 2.10. OPC Viability and Apoptosis

The influence of mice and human lymphocytes on OPC viability was assessed using three independent methods, which allowed analysis of OPC morphology, apoptosis/necrosis, and cell lysis.

#### 2.10.1. DIC Microscopy

The visualization of live cell interactions between lymphocytes and OPCs was performed on eight-well glass chamber slides (Nalge Nunc International, Waltham, MA, USA). During the course of 21 h, lymphocyte–OPC interactions were imaged at five time points (0, 1, 3, 5, and 7 h intervals) with a Zeiss Axiovert 200 inverse microscope with a Zeiss LD Plan-Neofluar 40×/0.62 Ph2 Korr differential interference contrast objective (Göttingen, Germany).

#### 2.10.2. OPC Apoptosis

The binding of ANXV-FITC to phosphatidylserine was used as a sensitive measurement of OPC apoptosis. Additional staining with (PI) enabled to distinguish between early and late stage of apoptosis, as ANXV binds to both types of cells. After incubation of OPCs with lymphocytes, samples (100 μL) were washed twice in ice cold PBS without Ca^+2^/Mg^+2^ and incubated with ANXV-FITC and PI (BD Pharmingen) according to the manufacturer’s instructions. OPCs were identified and gated on the SSC/FSC dot plot and analyzed by flow cytometry.

#### 2.10.3. Lactate Dehydrogenase (LDH) Release Assay

LDH release was measured by colorimetric method using Cytotoxicity Detection Kit (Sigma-Aldrich, St. Louis, MO, USA) according to the manufacturer’s instructions. The concentration of released LDH from 100% cell lysis (OLs lysed with 1% Triton X-100) was used as the positive control and supernatants from OLs supplemented with 10% PBS as negative control (spontaneous LDH release). The rate of lysed cells was calculated based on the normalization of each sample to the level of LDH released by positive control subtracted from negative control samples. All samples were analyzed in duplicate.

### 2.11. ELISA

Human BDNF, CNTF, PDGF subunit A, PDGF-Rα, FGF2, IGF-1 (all from Wuhan EIAab Science, Wuhan, China), sCD40 (Biorbyt, Cambridge, UK), MBP, PLP, and MOG (all from Biomatik, Wilmington, DE, USA) were measured in supernatants using ELISA Kits. The detection limits were as following: 0.33 pg/mL (BDNF and CNTF), 12 pg/mL (PDGFA and IGF-1), 0.045 ng/mL (PDGF-Rα), 1.56 pg/mL (FGF-2), 1.0 pg/mL (sCD40), 1.25 ng/mL (MBP), 0.056 ng/mL (PLP), and 0.054 ng/mL for MOG.

### 2.12. Immunocytochemical Analysis (ICC)

For ICC analysis, isolated cells were transferred to gelatin-coated microscope slides by cytospin (300× *g*, 10 min) and fixed with 4% PFA for 20 min at 21 °C. Fixed cells were washed with PBS and blocked with 10% rabbit blocking serum (Santa Cruz Biotechnology, Dallas, TX, USA) supplemented with 3% Triton^TM^ X-100 (Sigma-Aldrich, St. Louis, MO, USA) for 45 min at 21 °C. Cells were washed and double stained for MBP/MOG, PLP/nestin, O4/phalloidin, GFAP/phalloidin and Ago2/Dicer1. Anti-MOG (mouse, Santa Cruz Biotechnology, Dallas, TX, USA), anti-O4 (clone O4, R&D Systems), anti-Nestin (rabbit, BioLegend, San Diego, CA, USA), anti-Dicer1 (clone F10, Santa Cruz Biotechnology, Dallas, TX, USA), phalloidin (F-actin)/TR (Invitrogen), anti-Ago2 (rabbit, Abcam, Cambridge, UK), anti-GFAP (clone H50, Santa Cruz Biotechnology, Dallas, TX, USA), and rat IgG2b (Invitrogen, Carlsbad, CA, USA) as negative isotype control, were used. All antibodies were suspended in PBS supplemented with 1.5% blocking rabbit serum, 0.3% Triton X-100, 0.01% sodium azide, and incubated overnight at 4 °C. Cells were washed and secondary fluorescent Abs were added for 1 h at RT: chicken pAb to rabbit TR (Santa Cruz Biotechnology, Dallas, TX, USA), goat pAbs to mouse FITC (Abcam, Cambridge, UK), goat pAbs to mouse TR (Abcam, Cambridge, UK), mouse to rabbit FITC (Santa Cruz Biotechnology, Dallas, TX, USA). For nuclei DNA staining, DAPI (1.5 μg/mL UltraCruz Mounting Medium, Santa Cruz Biotechnology, Dallas, TX, USA) was used. Images were acquired using confocal microscope Nikon D-Eclipse C1 and analyzed with EZ-C1 v. 3.6 software. Fluorescence intensity was determined as the average fluorescence (Avg. area), the sum of the fluorescence from all segments divided by the number of segments. The average fluorescence was calculated using at least twenty single cells taken from four independent experiments. The level of baseline fluorescence was established individually for each experiment. Nonspecific fluorescence (signal noise) was electronically diminished to the level when nonspecific signal was undetectable [16].

### 2.13. CLARITY Brain Examination

CLARITY (Clear, Lipid-exchanged, Acrylamide-hybridized Rigid, Imaging/immunostaining compatible, Tissue hYdrogel) was performed according to RapiClear 1.55 Kit manufacturer’s instructions (SunJin Lab, Hsinchu City, Taiwan). Mice were anesthetized and perfused with ice-cold PBS (10 mL/min) and 4% PFA (10 mL/min). Brains and spinal cords were removed, placed into 20 mL of 4% PFA and incubated at 4 °C overnight. Next, brains and spinal cords were washed with PBS and transferred to 20% sucrose/PBS (*w*/*v*) for one day at 4 °C. Samples were placed into OCT Compound solution (Sakura Finetek, Torrance, CA, USA INC) and frozen overnight. Brains and spinal cords were washed with PBS, fixed with 4% PFA for 2 h at RT, washed again and treated with SCALEVIEW-A2 solution (Olympus Life Science) for two days. Subsequently, samples were transferred into ScaleB4 solution (8 M Urea with 0.1% Triton X-100) for 10 days being refreshed every 2–3 days and to working solution prepared by 3:2 diluting RapiClear^®^ 1.55 with ScaleB4 for 4–7 days. After the CLARITY procedure, samples were prepared for immunofluorescence analysis. Primary Abs anti-ELL3 (rabbit, Bioss Antibodies, Woburn, MA, USA), Exportin 5 (clone A11, Santa Cruz Biotechnology, Dallas, TX, USA), and MBP (chicken, ThermoFisher Scientific, Waltham, MA, USA) were applied for 96 h at 37 °C, then washed with buffer containing 0.5 M boric acid with 0.1% Triton X-100 (pH 8.5) for 48 h. Secondary Abs goat antichicken/TR (ThermoFisher Scientific, Waltham, MA, USA), goat antimouse/AlexaFluorPlus 488 (ThermoFisher Scientific, Waltham, MA, USA), goat antirabbit/AlexaFluor 350 (ThermoFisher Scientific, Waltham, MA, USA) were applied at 1:300 each for 72 h at 37 °C, and rinsed several times over a 96 h period. At the time of imaging, tissues were transferred into imaging spacer (iSpacer, SunJin Lab, Hsinchu City, Taiwan) and immersed in working solution. Images were acquired using a Leica TCS SP8 confocal microscope with 25× CLARITY objective.

### 2.14. Total RNA Isolation and miRNA Concentration Analysis

Total RNA was extracted from human and mouse OPCs and OLs with mirVanaTM miRNA Kit (ThermoFisher Scientific, Waltham, MA, USA). After isolation, the miRNA level and the purity analysis were performed by Agilent small RNA Kit (2100 Bioanalyzer, Agilent 2100 expert software, Santa Clara, CA, USA). The data were shown as raw data of miRNA concentration (pg/mL) as well as the percentage of small RNA.

### 2.15. Microarray and Data Bioinformatic Analysis

RNA integrity number (RIN) was measured by Agilent Bioanalyzer technology to check RNA quality. Specific mRNA expression was analyzed using Affymetrix^®^ GeneChip^®^ AF-902120 Affymetrix Mouse Gene 2.1 ST microarray expression profiling. The hybridization data were analyzed using Transcriptome Analysis Console (TAC) Software (ThermoFisher Scientific, Waltham, MA, USA).

### 2.16. Quantitative Real-time PCR (qRT-PCR)

qRT-PCR was performed with TaqMan probes (ThermoFisher Scientific, Waltham, MA, USA) using a 7900 Real Time PCR System. A total of 1 µg of total RNA was transcribed to cDNA using SuperScript^®^ VILO^TM^ cDNA Synthesis Kit (ThermoFisher Scientific, Waltham, MA, USA). cDNA was amplified in the presence of specific TaqMan probes: MBP (UniGene: Hs00921945_m1), MOG (Hs01555268_m1), PLP (Hs00166914_m1), Dicer1 (Hs00229023_m1), Ago2 (Hs01085579_m1), ELL3 (Hs00228559_m1), Exp5 (Hs00382453_m1), β-actin (Hs00921945_g1) using TaqMan^®^ Fast Advanced Master Mix (ThermoFisher Scientific, Waltham, MA, USA). The PCR reactions were performed on the 96-well plates using TaqMan^®^ Fast Advanced Master Mix (ThermoFisher Scientific, Waltham, MA, USA). The reaction specificity was checked by melting curve analysis, and relative gene expression was determined by the ΔΔCT method.

### 2.17. miRNA Library Preparation and Next Generation Sequencing (NGS)

miRNA profiling of OLs was performed using NGS. The library preparation was done using the NEBNext^®^ Small RNA Library preparation kit (New England Biolabs, Ipswich, MA, USA). A total of 500 ng of total RNA isolated from OLs in each independent experiment was converted into the miRNA NGS library. Library preparation quality control was performed using Bioanalyzer 2100 and TapeStation 4200 (both Agilent). Based on the quality of the inserts and the concentration measurements, the libraries were pooled in equimolar ratios. The pool was size-selected using the LabChip XT (PerkinElmer, Waltham, MA, USA), aiming to select the fraction with the size corresponding to miRNA libraries (~145 nt). The library pools were quantified using the qPCR KAPA Library Quantification Kit (KAPA Biosystems, Indianapolis, IN, USA). Samples were analyzed on a NextSeq 500. Raw data was demultiplexed and FASTQ files for each sample were generated using the bcl2fastq software (Illumina Inc., San Diego, CA, USA) [17].

### 2.18. Bioinformatic Analysis of NGS miRNA Data

miRNA bioinformatic data analysis was performed using reference annotation in homo sapiens GRCh37 and miRbase20 (http://www.mirbase.org/). All samples passed quality control (higher Q-score than Q30). Mapping of the sequencing reads were classified as follows: 1. Out-mapped—not used in the analysis (polyA and PolyC homopolymers, ribosomal RNA mitochondrial chromosome, and the genome of phiX174); 2. Unmapped—reads that could not be alignment to reference genome; 3. Genome—reads that aligned to the reference genome in locations with unknown miRNAs or smallRNAs, including fragments of mRNA and lncRNA transcripts; 4. miRNA—reads mapped to miRBase; 5. smallRNA—reads mapped to smallRNA database; 6. predicted miRNA—prediction based on the sequence homology in another organism or by miRPara predictive algorithm [18]. After mapping the data, counts of relevant entries in miRBase20 and the numbers of known microRNAs were calculated. The differential expression analysis was done using M-values normalization method (TMM normalization) by EdgeR statistical software package (Bioconductor, http://www.bioconductor.org/) [19]. NGS analysis was performed using Exiqon platform (Vedbaek, Denmark).

### 2.19. miRNA Target Analysis

Bioinformatics identification of hsa-miR-665, hsa-miR-21-3p, and hsa-miR-212-3p putative mRNA target sequences and functional classification of encoded proteins were performed based on a cooperative analysis of two predictive programs: miRSearch (www.exiqon.com/miRSearch) and miRBase (www.mirbase.org). The mirSVR score was used to classify the most important targets of miRNA which had the strongest affinity and the best matching to particular mRNA.

### 2.20. Validation of OL miRNA Expression with the Dropped Digital PCR System

miRNA quantification of miR-665, hsa-miR-21-3p, and hsa-miR-212-3p in OLs was done using specific TaqMan gene expression probes (UniGene Hs06637014_s1 for miR-665, 002438 for hsa-miR-21-3p and 000515 for hsa-miR-212-3p; all from Life Technologies, Carlsbad, USA) and the QX200 dropped digital PCR system (ddPCR, BioRad QuantaSoft Analysis Pro v.1.0.596 software, Hercules, CA, USA). ddPCR reactions were performed from additional cohort patients. The results were expressed as the number of miRNA copies per million cells.

### 2.21. Statistics

Arithmetic means and standard deviations were calculated for all parameters from at least four independent experiments. Statistical analysis of difference was performed using the one-way ANOVA test. Tukey’s test was used for multiple comparisons as a post-hoc test when statistical significances were identified in the ANOVA test. Statistical significance was set at *p* < 0.05.

## 3. Results

### 3.1. Coculture of RR-MS CD49d^+^CD154^+^ Lymphocytes Together with Maturing OPCs Results in Shifting Growth Factor Profile

We previously showed that interaction of RR-MS CD49d^+^CD154^+^ lymphocytes with maturating human oligodendrocyte precursor cells (hOPCs) resulted in increased production of proinflammatory cytokines/chemokines [10]. Therefore, we concluded that autoreactive lymphocytes might also affect the synthesis of growth factors critical for remyelination. To address this issue, we analyzed concentrations of brain-derived neurotrophic factor (BDNF), ciliary neurotrophic factor (CNTF), fibroblast growth factor (FGF-2), insulin-like growth factor-1 (IGF-1), platelet derived growth factor subunit A (PDGF-A), and platelet-derived growth factor receptor A (PDGFR-α) in the coculture supernatants of sorted RR-MS CD49d^+^CD154^+^ lymphocytes with maturing hOPCs. Both populations produced growth factors; RR-MS PBMCs in response to myelin peptides opposite to HC PBMCs produced: BDNF, CNTF, and PDGF-A, while hOPCs synthetized BDNF and IGF-1 during maturation. In the coculture of hOPCs with RR-MS CD49d^+^CD154^+^ lymphocytes, significantly higher concentrations of IGF-1, PDGF-A, and PDGF-R-α were detected in comparison to coculture with HC CD49d^+^CD154^+^ lymphocytes, as seen in Table 1.

### 3.2. RR-MS CD49d^+^CD154^+^ Lymphocytes and EAE Mouse Brain-Infiltrating CD3^+^/CD19^+^ Memory Cells Affect Maturing OPCs, Resulting in Dysregulation of Myelin Production by Mature OLs

As RR-MS CD49d^+^CD154^+^ lymphocytes produced growth factors affecting the remyelinating environment, we hypothesized that uniform thin and shortened internodes of axons seen in the remyelinating areas can result from disproportional MOG/PLP/MBP synthesis by mature OLs. We found a simultaneous increase in MBP (0.14 ± 0.064 ng/mL HC vs. 0.23 ± 0.082 RR-MS) and decrease in PLP (5.7 ± 0.58 vs. 3.1 ± 0.72) production by mature OLs in the presence of RR-MS myelin-specific CD49d^+^CD154^+^ lymphocytes, while MOG synthesis was not altered, as seen in Figure 1A. MBP/PLP index even more clearly exhibited disproportion in myelin protein synthesis by OLs in the presence of RR-MS CD49d^+^CD154^+^ lymphocytes both at mRNA (0.9 ± 0.17 HC vs. 9.3 ± 1.93 RR-MS) and protein levels (0.02 ± 0.003 HC vs. 0.07 ± 0.012 RR-MS), as seen in Figure 1B. ICC analysis supports these findings, as the most visible changes were observed in MBP and PLP intensity, as seen in Figure 1C. However, this effect was not caused by cytopathic impact of lymphocytes on hOPCs or disturbance in their maturation. ICC analysis revealed no changes in phalloidin/O4 (OLs mature marker), phalloidin/GFAP (astrocyte marker), or nestin expression (OPC marker) in hOPCs incubated with RR-MS or HC CD49d^+^CD154^+^ lymphocytes, in comparison to hOPCs incubated alone, as seen in Figure 1C. RR-MS CD49d^+^CD154^+^ lymphocytes did not also cause OL lysis, as demonstrated by LDH assay, as seen in Appendix A. Moreover, we observed a decreased rate of hOPC apoptosis in the presence of RR-MS CD49d^+^CD154^+^, while HC CD49d^+^CD154^+^ lymphocytes had no effect on apoptosis or LDH levels at all, as seen in Appendix A.

To confirm whether our in vitro observations are reflected at the site of remyelinating plaques in vivo, we used the mouse model of MS (EAE). As RR-MS CD49d^+^CD154^+^ lymphocytes induced in the periphery consisted of both T and B cells, we hypothesized that these subpopulations can cooperate in deregulation of myelin synthesis by OLs [10]. Therefore, lymphocytes isolated from EAE brains were sorted into T (CD3^+^) and memory B (mCD19^+^) cells and then cocultured with OPCs isolated from healthy newborn mice (mOPC), as seen in Figure 2A. Upon isolation of lymphocytes, they were mixed at a ratio of 10:1 (CD3^+^/mCD19^+^) to reflect the results of chemotaxis experiment of human RR-MS CD49d^+^CD154^+^ lymphocytes [10]. Due to the limited cell number, we used the ICC technique to confirm the role of EAE CD3^+^/mCD19^+^ BMC-derived cells in dysregulation of myelin synthesis by mOLs. Both ICC and quantitative mRNA analysis demonstrated high MBP and low PLP expression, while MOG levels were not affected during the coculture of mOPCs with CD3^+^/mCD19^+^ BMC-derived cells, as seen in Figure 2B,C. Although T cells alone were able to dysregulate myelin synthesis in OLs, the most visible differences were noted in the cocultures where both CD3^+^ and mCD19^+^ were present. CD3^+^/mCD19^+^ BMC-derived cells did not affect the morphology of maturing OPCs or their viability/apoptosis. DIC microscopy, as seen in Appendix A, and ICC analysis revealed no changes in phalloidin/O4 (OLs mature marker), phalloidin/GFAP (astrocyte marker), or nestin expression (OPC marker) in mOPCs incubated with EAE CD3^+^ or mCD19^+^ cells, in comparison to mOPCs incubated alone, as seen in Figure 2B. To confirm that dysregulated protein synthesis was not a result of OL lysis, LDH assay and ANX-V/PI labelling were performed. Similar to human cells, CD3^+^/mCD19^+^ BMC-derived cells did not induce lysis or apoptosis of mouse maturing OPCs, as seen in Appendix A.

### 3.3. EAE BMC-Derived Cells as Well as Human RR-MS CD49d^+^CD154^+^ Lymphocytes Have No Direct Effect on the OPC Transcriptional Factors, but Affect miRNA Synthesis

Maturation/proliferation of OPCs, and, in consequence, myelin synthesis, as crucial elements of regenerative processes, are regulated by the expression and activity of several transcriptional factors. TCF7L2 (Transcription factor 7-like 2) and SOX2 (Sex determining region Y-box 2) are two transcriptional factors critical for initial activation and increased proliferation of OPCs in response to tissue injury, whereas SFMBT2 (Scm-like with four MBT domains protein 2), MYC proto-oncogene protein, and transcription factor F2F1 control OPC differentiation into mature, myelin-forming OLs [20]. We confirmed an increased mRNA expression of these factors during mOPC maturation (relative fold change compared to nonpolarized cells). No statistically significant mRNA changes were detected when mOPCs were cultured with EAE BMC-derived CD3^+^, mCD19^+^, or CD3^+^/mCD19^+^, as seen in Figure 3A).

Several studies have emphasized that miRNA molecules which act on posttranscriptional level play an important role in controlling lineage-specific development and functioning of various cell types [21]. RR-MS, contrary to HC CD49d^+^CD154^+^ lymphocytes, decreased intracellular miRNA concentrations in hOPCs and diminished relative proportions of miRNA to small RNA, as seen in Figure 3B, right panel. We demonstrated similar results in mOPCs and EAE BMC-derived T cells and memory B cells. Specifically, CD3^+^ and mCD19^+^ cells at a ratio of 10:1 had greater effect on miRNA concentrations than CD3^+^ cells alone. mCD19^+^ cells alone did not affect the level or proportion of miRNA/small RNA in mature OLs, as seen in Figure 3B, left panel. Significant reduction of miRNAs in OPCs after exposure to proinflammatory cells can be a result of their synthesis in the nucleus and/or mechanisms regulating their cytoplasmic presence. miRNA biogenesis is a multistep process initiated by RNA polymerase II [22]. Using a microarray technique, we analyzed the expression of all known factors associated with the synthesis of miRNA and its intracellular activity in mOPCs. A bioinformatic analysis of mOPCs cultured with EAE brain-infiltrating CD3^+^, mCD19^+^, or CD3^+^/mCD19^+^ cells indicated 43 genes characterized by the highest fluctuations. Twenty genes were downregulated and twenty-three were upregulated in mOPCs in the coculture with EAE brain-infiltrating CD3^+^/mCD19^+^ cells. However, statistically significant differences were observed only in ELL3 (an RNA polymerase II elongation factor), exportin 5 (Exp5), DICER1, and Ago2, which were downregulated in comparison to mOPCs cultured alone, as seen in Figure 3C,D. We did not notice any changes in the expression of other proteins directly involved in miRNA processing, DGCR8, DROSHA, GW182, Ago1, 3, 4, and FXR1, as seen in Figure 3D. Validation of ELL3 and Exp5, the two main factors responsible for synthesis and transporting immature miRNA from nucleus to cytoplasm at the protein level, was done with the quantitative protein analysis as a whole brain imaging by CLARITY. Using healthy and EAE mouse brains collected at the peak of disease (3.5–4 score) and three weeks after the acute phase (1–1.5 score), we performed an analysis of ELL3, Exp5, and MBP expression in the gray matter and demyelinating plaques, as seen in Figure 4A. During the acute phase, we noted that at the sites characterized by low MBP level (demyelinating plaques), ELL3 and Exp5 expressions were significantly higher than three weeks after the peak of the disease, as seen in Figure 4B. During the remyelinating phase (three weeks after the peak of disease), we observed inversed expression of these proteins: MBP fluorescent signal was high, whereas ELL3 and Exp5 reduced. Additionally, high overlap coefficient (r^2^ > 0.89 for ELL3-MBP and EXP5-MBP, *p* < 0.05) pointed to a strong correlation between low ELL3/Exp5 expression and high MBP synthesis during remyelination and on the contrary, high ELL3/Exp5 and low MBP expression at the peak of disease (r^2^ > 0.97 for ELL3-MBP, 0.92 EXP5-MBP, respectively, *p* < 0.05), as seen in Figure 4B, upper-right panel.

Several studies have demonstrated that miRNA concentrations in cytoplasm are related to DICER1 and Argonauts protein expression, and vice versa, the concentrations of these proteins were dependent on miRNA levels [16,23,24]. To find out if such an association exists in our study, we used the ICC method to analyze cytoplasmic proteins involved in miRNA processing. DICER1 and Ago2 proteins were significantly downregulated in mOPCs cocultured with EAE BMC-derived CD3^+^/mCD19^+^ cells, as seen in Appendix A, which confirmed the observations detected at the mRNA level.

### 3.4. RR-MS CD49d^+^CD154^+^ Lymphocytes Interfere with miRNA Profiling in Maturing OPCs

To explore how dysregulation in miRNA protein compositions affects particular miRNA production in maturing OPCs exposed to RR-MS CD49d^+^CD154^+^ lymphocytes, we analyzed miRNA expression using NGS. Because different miRNA molecules target the same mRNA target in mice and humans, we used hOPCs as the most adequate model to draw the conclusions. We selected fifty miRNAs expressed in hOPCs with the highest coefficient of variation (%CV). A heat map diagram revealed that each of four examined hOPC cultures had a specific miRNA profile, as the results obtained from independent samples in each group were homogenous and formed clusters, as seen in Figure 5A, left panel). Taking into account the changes in gene expression of hOPCs cultured in different conditions, we have identified three clusters. The first one contained hsa-miR-21-5p, hsa-miR-221-5p, hsa-miR-222-3p, hsa-miR-21-3p, and hsa-miR-665, which were upregulated in the presence of RR-MS CD49d^+^CD154^+^ lymphocytes, as seen in Figure 5A, red line. The unique role of RR-MS CD49d^+^CD154^+^ lymphocytes in this cluster was emphasized by the fact that no differences were detected in hOPCs cultured with HC CD49d^+^CD154^+^ lymphocytes or in MO3.13 cells not differentiated vs. hOPCs. The second cluster contained hsa-miR-4485, hsa-miR-36-7-3p, hsa-miR-135b-5p, hsa-miR-4454, hsa-miR-337-5p, and hsa-mir-548ah-3p, which were downregulated in the presence of RR-MS CD49d^+^CD154^+^ lymphocytes, while no changes were demonstrated in the presence of HC CD49d^+^CD154^+^ lymphocytes compared to hOPCs, as seen in Figure 5A, green line. The third one was characterized by upregulation of hsa-miR-5701, hsa-miR-212-3p, and hsa-miR-1248 after exposition to HC but not RR-MS CD49d^+^CD154^+^ lymphocytes, as seen in Figure 5A, blue line.

Finally, using volcano plot system analysis, we selected molecules with the highest fold difference in the expression between cocultures of OPCs with RR-MS CD49d^+^CD154^+^ lymphocytes and HC CD49d^+^CD154^+^ lymphocytes. The most significant differences were observed in hsa-miR-21-3p, hsa-miR-665 which were upregulated, and hsa-miR-212-3p which was downregulated, as seen in Appendix A. These results were next validated by ddPCR in a separate cohort of four RR-MS patients and five HCs. The highest concentration of hsa-miR-21-3p and hsa-miR-665 in hOPCs was observed in the presence of RR-MS CD49d^+^CD154^+^ lymphocytes, as seen in Figure 5B. The diminished concentration of miR-212-3p in hOPCs was observed only in the presence of RR-MS CD49d^+^CD154^+^ lymphocytes. HC CD49d^+^CD154^+^ lymphocytes did not affect the concentration of any tested miRNAs.

### 3.5. The Crucial Role of miR-665 in Dysregulation of Myelin Protein Synthesis in hOPCs Exposed to RR-MS CD49d^+^CD154^+^ Lymphocytes

Bioinformatic analysis of the potential effect of miR-21-3p, hsa-miR-665, and miR-212-3p on protein-coding transcripts revealed that none of these miRNA molecules were directly involved in myelin mRNA protein regulation. Twenty-two selected miRNAs with the highest mirSVR scores encode proteins belonging to transcriptional and DNA-interacting factors, which are responsible for gene regulation, as seen in Appendix A. Among these factors, the highest mirSVR rate was assigned to BTAF1 RNA polymerase II (−2.30 mirSVR score), which is negatively regulated by hsa-miR-665. We investigated whether upregulated hsa-miR-665 and miR-21-3p can indirectly influence the expression of genes necessary for myelin protein synthesis. We used antisense hsa-miR-21-3p and hsa-miR-665 transfected inhibitors to confirm their role in modulation of ELL3 and myelin protein transcripts as well as total miRNA intracellular concentration using hOPC model, as seen in Figure 6A. Similar to the observations made in mOPCs, in hOPCs, we also found decreased expression of ELL3 in cocultures with RR-MS CD49d^+^CD154^+^ lymphocytes, as seen in Figure 6B, miR negative control. Neutralization of hsa-miR-665 not only inhibited this effect but caused the increase of ELL3 mRNA expression. Additionally, we noted that in the cocultures of CD49d^+^CD154^+^ lymphocytes with hOPCs and neutralizing hsa-miR-665, the extent of the previously observed increase of MBP expression became reduced, the PLP mRNA was downregulated, and MOG mRNA expression was constant, as seen in Figure 6B. Anti-hsa-miR-21-3p did not affect ELL3, PLP, or MOG mRNAs expression. Although anti-hsa-miR-21-3p reduced MBP mRNA expression, this impact was not as potent as during miR-665 neutralization. Anti-hsa-miR-665, but not anti-hsa-miR-21-3p, led to total miRNA intracellular normalization, as seen in Figure 6C.

## 4. Discussion

It is generally hypothesized that limited remyelination after MS relapse is associated with the incapability of OPCs to repopulate the area of demyelination, which is the effect of inhibitory factors and/or lack of stimuli required to generate remyelinating OLs in the area of demyelination [25].

Our study provides the detailed explanation about the causes of inefficient production of myelin proteins. This phenomenon is related to the presence of brain-infiltrating myelin-specific CD49d^+^CD154^+^ lymphocytes in remyelinating zone during MS remission. Maturing OPCs are reprogrammed into immune reactive OLs that cannot efficiently produce myelin proteins, but instead of this, amplify recruitment and mediate proliferation of myelin-specific lymphocytes as a result of two different mechanisms. Firstly, these lymphocytes might migrate from the periphery, recruited by chemokines released within remyelinating plaque by maturating OPCs, and are intensely accelerated by interaction with immune-reactive OLs. Secondly, myelin-specific lymphocytes can proliferate following direct contact with myelin antigens presented by maturating OPCs, leading to the development of the resident RR-MS CD49d^+^CD154^+^ lymphocytes, which are not completely eliminated after active phase of disease and which can permanently affect the remyelination process.

Our previous data demonstrated that RR-MS CD49d^+^CD154^+^ lymphocytes are divided into CD3 and CD19 subpopulations (10:1) [10]. As both populations of CD19^+^ and CD3^+^ lymphocytes were strongly attracted by maturating OPCs, we investigated their particular roles in the induction of reactive OLs. Specifically, we showed that EAE CD3^+^ and memory CD19^+^ (mCD19^+^) BMCs isolated during the remyelination phase cooperated in affecting MPB, PLP, and miRNA processing of protein production in OLs to a greater degree than CD3^+^ alone. Moreover, mCD19^+^ cells alone did not affect maturing OPCs at all, suggesting their influence on OPCs via supporting CD3^+^. This is in concordance with previous data demonstrating HLA-DR-dependent autoproliferation of self-reactive brain-homing CD4^+^ mediated by mCD19^+^ cells [26].

RR-MS CD49d^+^CD154^+^ cells affected the ratio of MBP/PLP produced by OLs and inhibited their spontaneous apoptosis. These changes are probably the effect of growth factors and cytokines/chemokines produced by RR-MS PBMCs exposed to myelin proteins. The role of these molecules in insufficient regeneration might be postulated, given that approximately 70% of MS lesions contain progenitor cells or premyelinating OLs that are characterized by the inability to differentiate to mature myelin-producing OLs. The other 30% of lesions contain only a few progenitors, suggesting inability to migrate towards the demyelinating zone [8]. During physiological CNS reconstitution, growth factors FGF, IGF, CNTF, and PDGF are synthetized by perivascular astrocytes, and/or residual microglia/macrophages in normal-appearing grey matter, orchestrating maturation of OPCs and neurons [9]. Presence of brain-infiltrating immune cells that produce growth factors can change this favorable environment. We revealed the high concentration of BDNF, CNTF, and PDGF-A in supernatants of RR-MS PBMCs cultured with myelin proteins. Our observations are in the line with previous studies demonstrating that EAE brain-infiltrating immune cells were characterized by increased PDGF-A synthesis [27], and OPCs recruited towards the chronic-active lesions expressed IGFR1, FGFR1, and PDGFR1 growth factor receptors [28]. Because OPCs produce BDNF, FGF-2, and IGF-1, the interpretation of the coculture results is problematic and the true source of these growth factors can only be speculated [29]. As myelin proteins did not stimulate RR-MS PBMCs to produce IGF-1, its high concentration in the coculture might be a result of increased synthesis by maturating OPCs exposed to RR-MS CD49d^+^CD154^+^ lymphocytes. In this set of experiments, we also noted high concentrations of PDGF-A and its receptor. Since PDGF-A is not produced by maturating OPCs, but by myelin-stimulated RR-MS PBMCs, lymphocytes are probably the source of PDGF-A in the coculture [30]. Increased shedding of PDGF-R-α in cocultures pointed to functional engagement of PDGF-A in modulation of maturating OPCs. Regardless of the growth factor source, a comparative analysis of OPC/RR-MS CD49d^+^CD154^+^ lymphocyte coculture confirmed the unique properties of MS lymphocytes in modifying the environment of remyelinating plaque.

Our data suggest that pathological deregulation of OPC maturation is executed at the post-transcriptional level, and is dependent on the miRNA activity. First, we analyzed the expression of transcriptional factors which are crucial for activation and polarization of adult OPCs in response to a brain tissue degeneration: TCF7L2 and SOX2 responsible for adult OPC activation; SFMBT2, MYC proto-oncogen protein, and E2F1 responsible for OPC differentiation and production of myelin proteins [31,32]. Using mRNA microarray analysis, we noted that none of them changed their expression in maturating OPCs cocultured with EAE BMC-derived CD3^+^ and mCD19^+^ cells. Therefore, we assumed that OPC–BMC interaction engaged post-transcriptional changes most likely dependent on the miRNA activity.

In fact, we demonstrated that maturing OPCs cocultured with RR-MS or EAE brain-derived lymphocytes were characterized by dysregulated miRNA synthesis. This seems to be a result of ELL3 downregulation, which controls elongation step of the miRNA synthesis by RNA polymerase II (pol II) [33]. It is worthy to emphasize that the elongation step regulated by ELL3 is thought to be a critical stage for the regulation of gene expression, especially in stimulating lineage differentiation of embryonic stem cells [34].

CLARITY analysis of ELL3, Exp5, and MBP proteins in the brains from EAE mice at the peak of the disease and three weeks later vs. healthy mouse brains confirmed our ex vivo findings. During remyelination, MBP was intensively synthetized, but ELL3 and Exp5 expression were reduced. Our experiments demonstrated a link between dysregulation of miRNA processing proteins at the early stages of miRNA synthesis and MBP overexpression. Downregulated miRNA synthesis was reflected by lower expression of miRNA biogenesis proteins. We found that downregulation of ELL3 was accompanied by low expression of Exp5, DICER1, and Ago2 (at mRNA and protein levels). Downregulation of Exp5 resulted in decreased DICER1 protein through a post-transcriptional mechanism that was associated with nuclear accumulation of DICER1 mRNA and its interaction with Exp5 mRNA [35]. Although the role of Ago2, as the limiting factor for miRNA biogenesis is well-known, its overexpression in humans leads to the increase of total mature miRNA [23,36,37], no data is available about the reversed effect—the total miRNA level on the Ago2 expression. The last question we addressed was about the direct cause of ELL3 downregulation in maturating OPCs exposed to lymphocytes present in the remyelinating plaques. Transcriptional miRNAs are controlled in a feedback loop reaction where transcriptional factors, that regulate specific miRNAs, are themselves the targets for those miRNAs [38]. NGS analysis of MO3.13 cells polarized to myelin-producing OLs cocultured with RR-MS CD49d^+^CD154^+^ lymphocytes identified three differentially expressed molecules: miR-665, miR-21-3p, and miR-212-3p. Two of them, miR-665 and miR-21-3p, are negatively and directly connected with activity of polymerase II and global chromatin remodeling. Consequently, we found that miR-665 overexpression affects other miRNA molecules by interfering with transcription, and exerts an indirect effect on myelin protein synthesis. Experiments performed using antagomirs confirmed bioinformatic suggestions that miRNA dysregulation in maturating OPCs cultured with RR-MS CD49d^+^CD154^+^ lymphocytes was dependent on the miR-665, but not miR-21-3p overexpression, as only miR-665 neutralization reconstituted total miRNA concentrations and ELL3, MBP, PLP mRNA levels. Among the most abundant miRNAs in the mature OLs, miR-219 is thought to be crucial in promoting OL differentiation, in part by direct targeting negative regulators of OL development such as PDGFRα, Sox6, and Hes5 [39]. Deletion of DICER in mature OLs using PLP-CreERT results in the dysregulation of miR-219 target gene Elov7 expression that is involved in lipid homeostasis [40]. Although, in our study, RR-MS as well as HC CD49d^+^CD154^+^ lymphocytes had no effect on miR-219, and other miRNA molecules involved in OPC maturation (miR-338, miR-138, miR-29, miR-32), we cannot exclude that they are engaged in impaired myelin synthesis by OLs [21,41]. Our doubts come from the miRNA NGS analysis, where each sample was normalized to an equal amount of cDNA. As total miRNA was diminished by silencing RNA polymerase II transcription activity, and more total RNA was taken to cDNA synthesis, many more miRNAs can appear to be downregulated. Nevertheless, this part of the study demonstrated a new possible mechanism, which involved not only particular miRNA molecules directly associated with myelin synthesis, but also affected the global miRNA profile at the transcriptional level. Our findings provide a link between disproportions in MBP and PLP production during remyelination via dysregulation of miRNAs in maturing OPCs triggered by RR-MS CD49d^+^CD154^+^ myelin-specific lymphocytes.

Our findings explain the phenomenon of insufficient remyelination based on the positive feedback between OPC-CD49d^+^CD154^+^ myelin-specific lymphocytes. Therefore, our data provide the opportunity for therapeutic interventions using combined therapy with neutralizing anti-CD154 and CD49d mAbs. As CD49d^+^CD154^+^ lymphocyte proliferation might be induced by OPCs in the CNS, the effectiveness of this therapy nowadays is limited. Further studies with the use of anti-CD154 mAbs alone, as well as in combination with anti-CD49d mAbs administrated intravenously and/or intraventricularly at different time points of EAE to estimate their influence on the disease induction and remyelination process may indicate a new trend for therapeutic approach.

## Figures and Tables

**Figure 1 cells-08-01508-f001:**
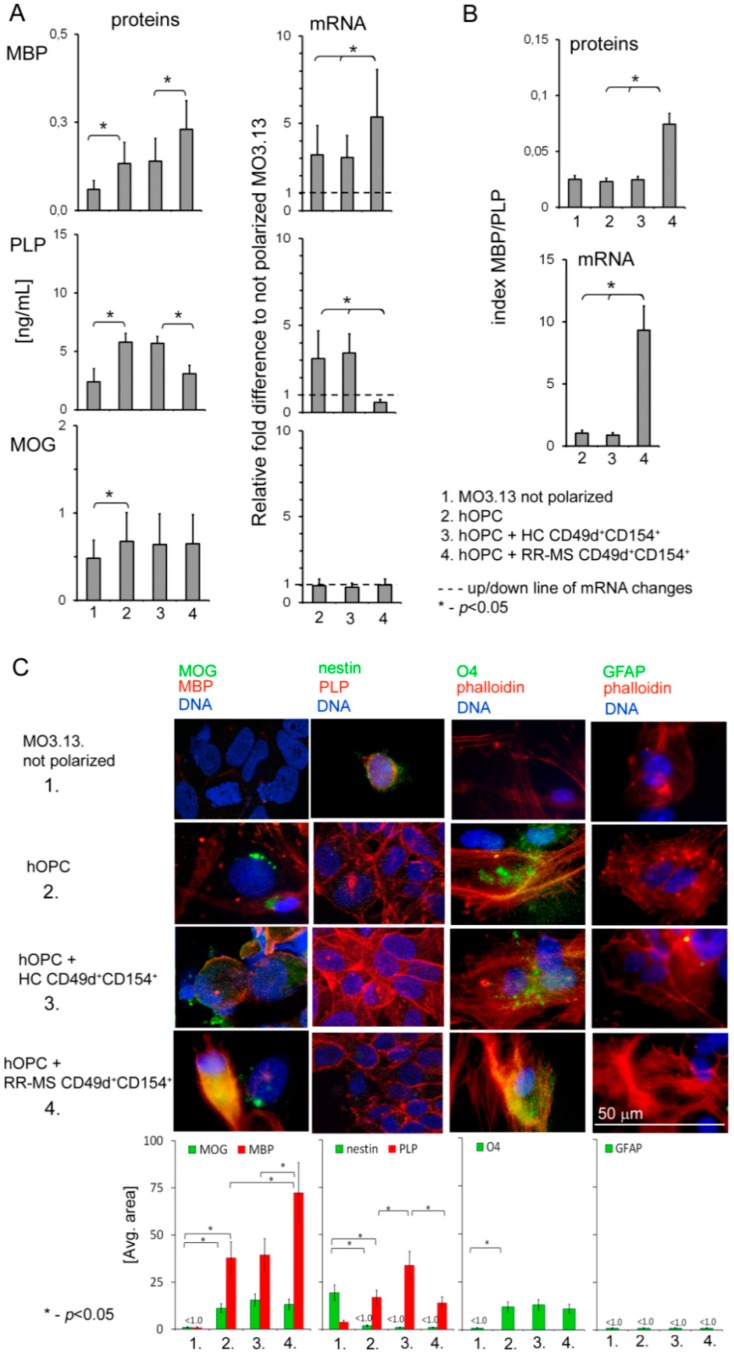
Relapsing-remitting multiple sclerosis (RR-MS) CD49d^+^CD154^+^ lymphocytes affect maturing oligodendrocyte precursor cells (OPCs), resulting in dysregulation of myelin production by mature oligodendrocytes (OLs). (**A**) RR-MS CD49d^+^CD154^+^, contrary to healthy control (HC) lymphocytes, affected myelin basic protein (MBP) and proteolipid protein (PLP) but not myelin oligodendrocyte glycoprotein (MOG) synthesis by human OPCs (hOPCs). (**B**) MBP/PLP index more clearly exhibited disproportion in myelin protein synthesis. (**C**) Immunocytochemical (ICC) analysis of nestin, O4 and GFAP expression in hOPCs demonstrated that RR-MS CD49d^+^CD154^+^ lymphocytes did not affect hOPC maturation, but caused dysregulation of MBP and PLP synthesis by mature OLs. All data are presented as means ± SD.

**Figure 2 cells-08-01508-f002:**
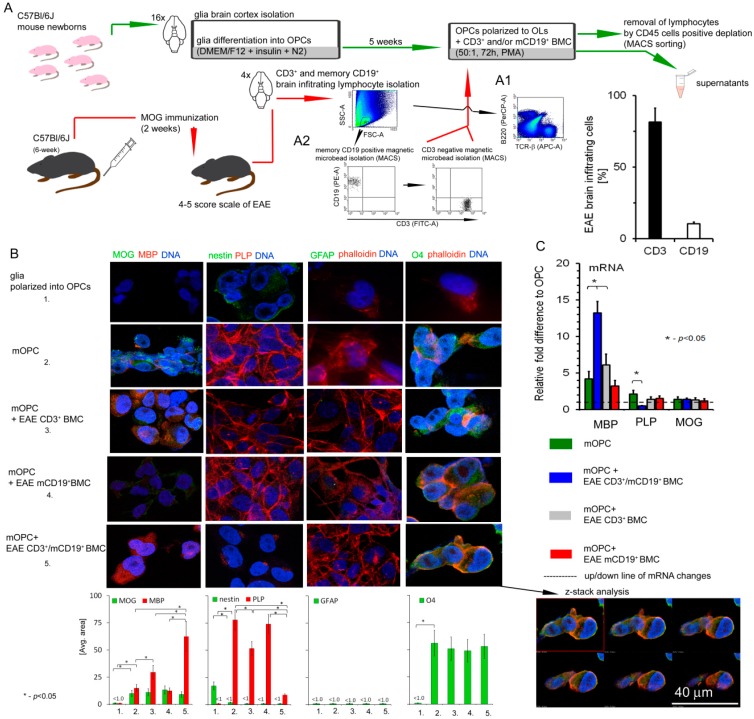
Mouse brain-infiltrating mononuclear cells (BMCs) affect myelin protein synthesis by mouse OPCs (mOPCs). (**A**) BMCs from Experimental Autoimmune Encephalomyelitis (EAE) mice in the third week after the peak of the disease (remyelination phase) were cocultured with OPCs isolated from the healthy mouse newborns (mOPCs). (A1, left panel) Phenotypic analysis of BMCs (region R1) divided into B cells (B220, region R2) and T cells (TCR-β, region R3). (A1, right panel) CD3^+^ and CD19^+^ cell rate analysis in isolated BMCs. (A2) Memory B cells (mCD19^+^) were positively selected, and unlabeled cells additionally sorted by negative selection into CD3^+^ lymphocytes from BMCs. (**B**, upper panel) In contrast to immature cells, mature OLs exhibited MBP/PLP/MOG expression (markers of the late stage of differentiation), O4 expression (marker of mature cells), and negative signal for nestin (marker of OL precursor cells) and GFAP (astrocyte-specific marker). (**B**, low panel) EAE CD3^+^/mCD19^+^ and CD3^+^ cells alone down- PLP and upregulated MBP expression, but did not affect O4 or nestin. Maturation process was associated with formation of actin-like microfilaments (labelled with phalloidin). Confocal z-stack analysis confirmed that EAE CD3^+^/mCD19^+^ BMC-derived cells did not affect O4 accumulation in the intracellular space and microfilament formation during OPC maturation. (**C**) mRNA myelin protein analysis proved that EAE CD3^+^/mCD19^+^ and CD3^+^ BMC-derived cells up- MBP and downregulated PLP, but had no influence on MOG synthesis. All data are presented as means ± SD.

**Figure 3 cells-08-01508-f003:**
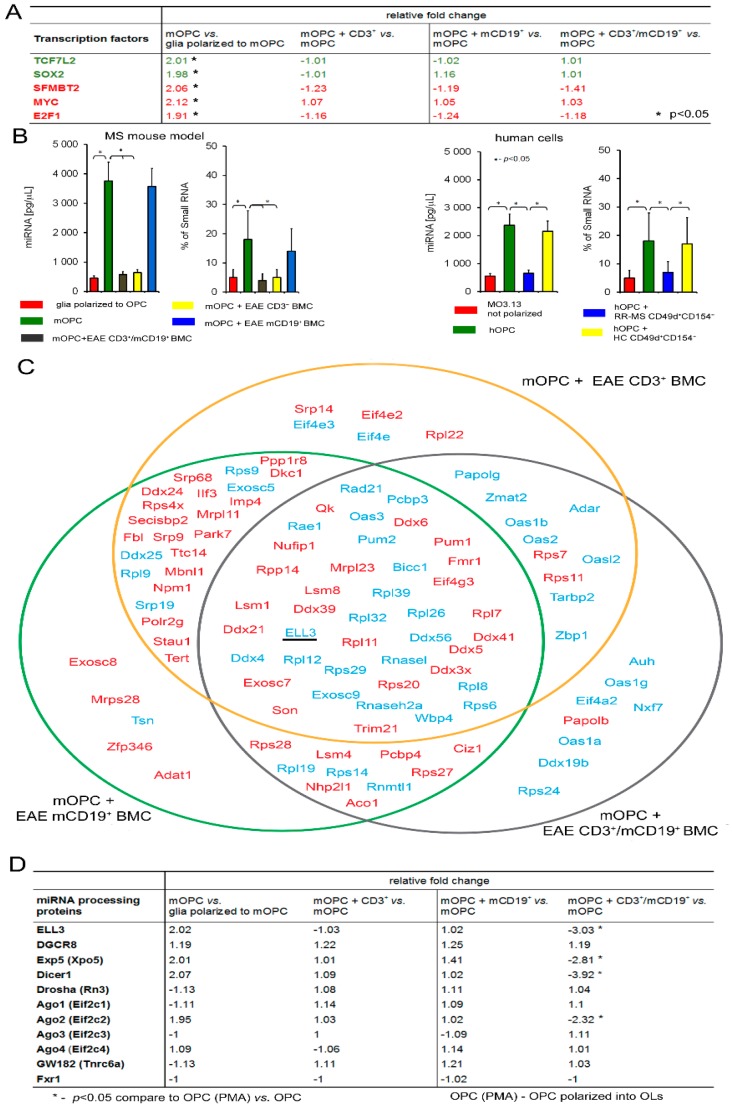
EAE BMC-derived cells as well as RR-MS CD49d^+^CD154^+^ lymphocytes affect miRNA synthesis, but have no direct effect on the transcriptional factors. (**A**) EAE BMC-derived cells did not change mRNA expression of transcription factors responsible for adult OPC activation (green font) as well as differentiation and myelin sheaths formation (red font). (**B**) RR-MS CD49d^+^CD154^+^ lymphocytes as well as CD3^+^/mCD19^+^ and CD3^+^ EAE BMC-derived cells affected intracellular miRNA concentrations during OPC maturation. Data are presented as means ± SD. (**C**) Bioinformatic analysis of genes responsible for mRNA binding and miRNA processing point to the polymerase II elongation factor (ELL3). From the total of 33,000 genes, those mostly affected during physiological OPC maturation and associated with mRNA binding or engagement in miRNA synthesis were selected. 43 genes from this group, with statistically significant fluctuations in the presence of CD3^+^/mCD19^+^ EAE BMC-derived cells were selected. Twenty-three were upregulated (red font) and 20 downregulated (blue font). The most significant alterations were observed in ELL3 expression. (**D**) The validation of ELL3 and the other miRNA processing proteins by qPCR points to Exp5, DICER1, and Ago2, which were reduced in maturating OPCs in the presence of CD3^+^/mCD19^+^ EAE BMC-derived cells.

**Figure 4 cells-08-01508-f004:**
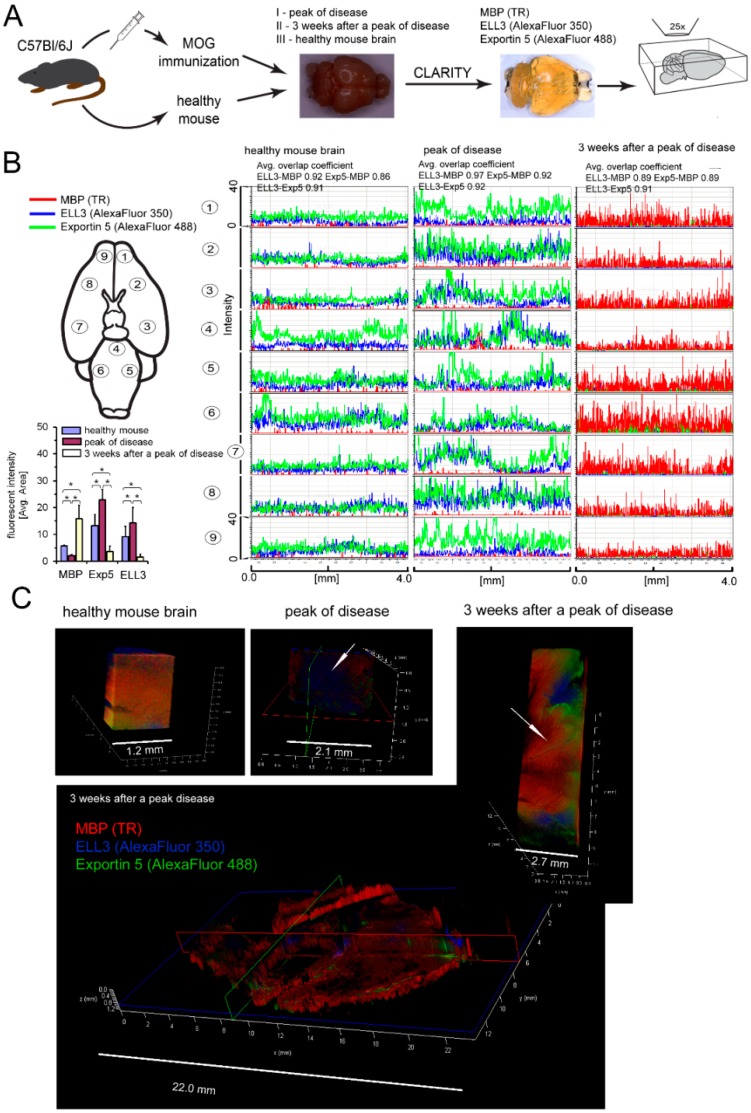
Pathological SM lesions in the period between the active and remission phase of disease characterized by changeable ELL3 and Exp5 expression which correspond with MBP synthesis. (**A**) Colocalization of ELL3 and Exp5 miRNA processing proteins in the region of the demyelinating and remyelinating plaques during the course of disease using the whole imaging of mouse EAE brain by CLARITY. (**B**, left upper panel) The fluorescent intensity of MBP (red pseudocolor), ELL3 (blue) and Exp5 (green) was measured in nine regions of each investigated brains in every 4.0 mm of tissue thickness. (**B**, left lower panel) Downregulation of ELL3 synthesis is associated with reduced Exp5 expression. Data are presented as means ± SD. (**B**, right panel) High colocalization of florescent signals (average overlap coefficients factor > 0.91) between MBP-ELL3, MBP-Exp5 and ELL3-Exp5 in all analyzed regions suggested that intensity of miRNA synthesis and its transport to cytoplasm was correlated with MBP expression. (**C**, left and middle pictures). In contrast to the healthy mouse, in the peak of EAE, demyelination regions were characterized by the loss of MBP and high ELL3 and Exp5 expressions (an arrow points demyelinating plaque). (**C**, lower and right pictures) After three weeks, this process was reversed as MBP was highly overexpressed and ELL3 and Exp5 were downregulated (an arrow points remyelinating region).

**Figure 5 cells-08-01508-f005:**
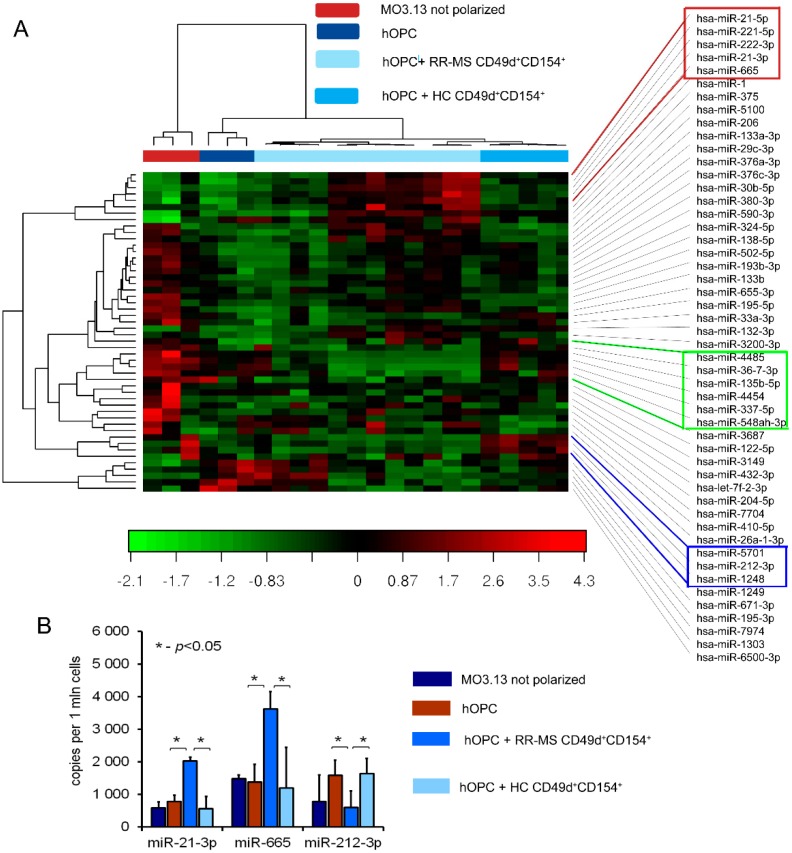
RR-MS myelin-specific CD49d^+^CD154^+^ lymphocytes interfere with miRNA profiling in maturing OPCs. (**A**). Heat map revealed a specific miRNA profile for every cell culture experimental settings. Based on the differences between the effect of RR-MS and HC CD49d^+^CD154^+^ lymphocytes on OPCs, three clusters were selected (red, green, and blue frames). The most significant differences were observed in the cluster with hsa-miR-21-5p, hsa-miR-221-5p, hsa-miR-222-3p, hsa-miR-21-3p hsa-miR-665 (red frame), and hsa-miR-212-3p (blue frame). The data was normalized with the trimmed mean of M-values method and converted to a log2 scale. Red represents an expression level above the mean, while green one level below. (**B**) Validation of selected miRNA by copy transcript estimation using ddPCR analysis in additional cohort patients. Data are presented as means ± SD.

**Figure 6 cells-08-01508-f006:**
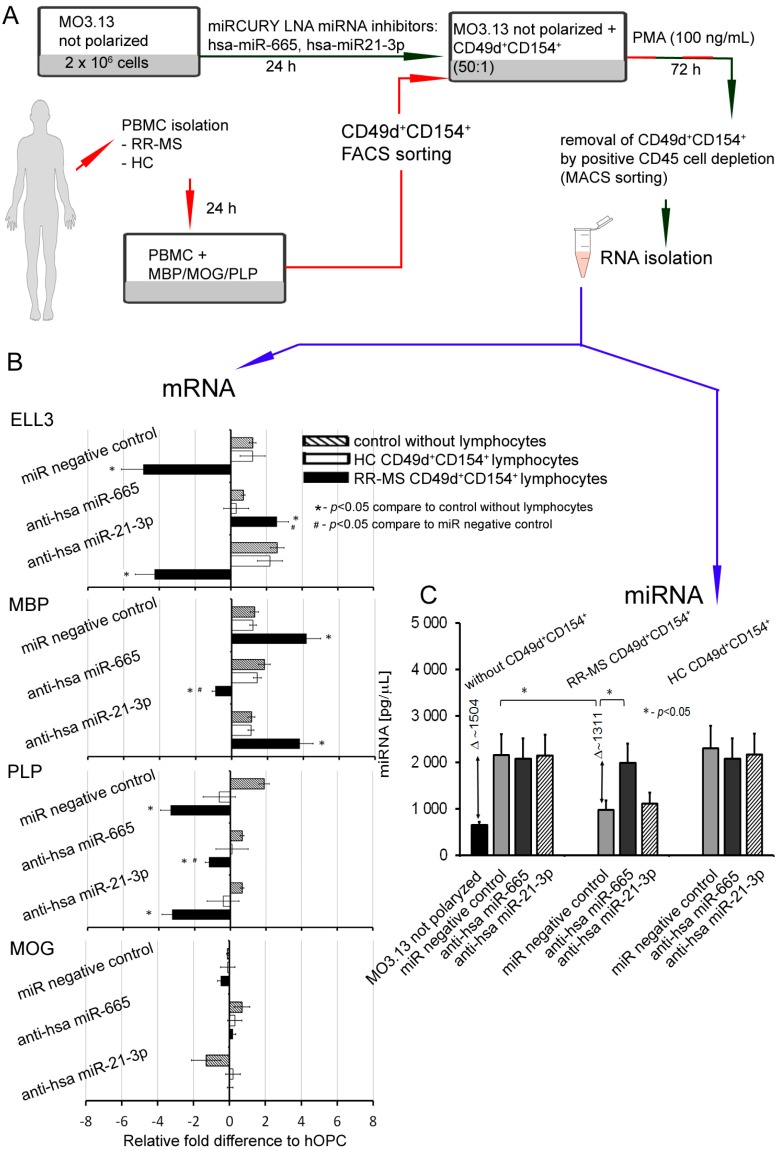
The crucial role of hsa-miR-665 and hsa-miR-21-3p in myelin protein dysregulations in hOPCs exposed to RR-MS CD49d^+^CD154^+^ lymphocytes. (**A**,**B**) Neutralization of hsa-miR-665 but not hsa-miR-21-3p, improved ELL3, PLP mRNA synthesis, and reduced MBP mRNA expression. (**C**) Neutralization of hsa-miR-665 but not hsa-miR-21-3p completely reversed the effect of RR-MS CD49d^+^CD154^+^ lymphocytes on miRNA synthesis by hOPCs. All data are presented as means ± SD.

**Table 1 cells-08-01508-t001:** RR-MS CD49d^+^CD154^+^ lymphocytes together with maturing OPCs affect the remyelination by disturbing environment created by growth factors. RR-MS PBMCs stimulated by MOG/PLP/MBP proteins opposite to HC produced BDNF, CNTF, and PDGF-A. RR-MS myelin-specific CD49d^+^CD154^+^ lymphocytes contrary to HC affect the IGF-1 synthesis by MO3.13 polarized to OLs (hOPCs). Data are presented as means ± SD.

	MO 3.13 (2 × 10^6^ cells/mL)				PBMC (2 × 10^6^ cells/mL)	
	Not Polarized	Polarized to OLs	Polarized to OLs + RR-MS CD49d^+^CD154^+^	Polarized to OLs + HC CD49d^+^CD154^+^	RR-MS	HC
	Growth factors [pg/mL]					
BDNF	<0.3	2.6 ± 0.67 ^†^	<0.3	<0.3	11.3 ± 3.92 ^#^	<0.3
CNTF	<0.3	<0.3	<0.3	<0.3	17.1 ± 8.95 ^#^	<0.3
FGF2	43.9 ± 12.09	40.7 ± 10.33	42.5 ± 12.79	44.1 ± 10.27	3.3 ± 0.98	2.1 ± 0.74
IGF-1	100.4 ± 39.55	157.6 ± 28.85 ^†^	279.5 ± 37.58 * ^§^	147.33 ± 28.71	21.7 ± 17.56	33.9 ± 10.94
PDGF-A	<12.0	<12.0	35.6 ± 15.19 * ^§^	<12.0	77.9 ± 11.19 ^#^	<12.0
PDGF-R-α (ng/mL)	<0.045	<0.045	3.9 ± 0.44 * ^§^	<0.045	<0.045	<0.045

^†^—statistically significant differences between MO3.13 and MO3.13 polarized to OLs; *—statistically significant differences between MO3.13 polarized to OLs and MO3.13 polarized to OLs with CD49d^+^CD154^+^ cells; ^#^—statistically significant differences between RR-MS and HC PBMC; ^§^—statistically significant differences between MO3.13 polarized to OLs with RR- MS CD49d^+^CD154^+^ and MO3.13 polarized to OLs with HC CD49d^+^CD154^+^ cells.

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
