# Peer review of "MS CD49d+CD154+ Lymphocytes Reprogram Oligodendrocytes into Immune Reactive Cells Affecting CNS Regeneration"

_cells, 2019, doi:10.3390/cells8121508_

Round 1

Reviewer 1 Report

This appears as a well written manuscript. I have a consideration about two issues.

Statistics: An unplanned comparison is a comparison made within a data set after an ANOVA test has been run, so the parameters of the comparison are not built into the ANOVA experiment. Scheffe's test can be used in situations where the results of an ANOVA experiment have yielded a significant F-statistic. This indicates that there is a meaningful difference in the means of the groups being compared. While Scheffe's test has the advantage of giving the experimenter the flexibility to test any comparisons that appear interesting, the drawback of this flexibility is that the test has very low statistical power. The authors must correctly explain why they used this post hoc test and not a Tukey's or a Bonferroni's test

qRT-PCR: The authors should specify if they used primers and which one.

Author Response

We agree with Reviewer’s opinion that Tukey test would be better as less conservative post-hoc test. Therefore, we recalculate the data of all experiments. Relevant corrections to the statistic description have been added. We specified primers by adding UniGene number in qRT-PCR analysis according to the Reviewer’s suggestion.

Reviewer 2 Report

Authors Piatek et al., submitted their manuscript titled "MS CD49d+CD154+ Lymphocytes Reprogram Oligodendrocytes Into Immune Reactive Cells Affecting CNS Regeneration" to Cells to be considered for publication. Authors using cell culture, imaging, biochemical and molecular biological techniques like CLARITY, qRT-PCR, miRNA library construction and analysis identified that CD49d+CD154+ lymphocytes potentially reprogram OPCs/ OL to activate inflammatory mechanisms and thus affecting remyelination in MS. Study presented by Piatek et al., is original and is of interest to the readers, I suggest the authors to revise the article according to my suggestions;

1) I would like to see major findings by Piatek et al., replicated in vivo. Using EAE model described here, is it possible to block or deplete CD49d+CD154+ lymphocytes? If so I recommend the authors to analyse in EAE-MOG model brains compared to controls a) MBP+ cells/ field, b) Olig2+ ki67+ cells/ field, and c) myelination index using staining and imaging techniques? Please use appropriate stats and mention in figure legends. Data from this study will significantly improve the applicability of this study presented here. Discuss this strategy in the manuscript.

2) Authors mentioned in methods section that they used female mice for experiments involving EAE models. Is there a valid reason to include only females? Please mention in methods section and discuss.

3) I am wondering about the influence of cell death and lysis of OPCs/OL that could influence this data. Can you please report cell death analysis using standard imaging or sorting techniques?

4) Images presented in Fig. 1C and 2B can be significantly improved to enhance quality. Nestin/ GFAP signal is not clear in those images presented. Also, quantification is needed for all these groups described in 1C and 2B. Please use appropriate stats and mention in figure legends.

5) In Fig. 2, next to 2A1, for EAE brain infiltrating cells (%)... it is not clear if these are % increase over control levels? Show control B6 values as baseline or include control values in graph. Please use appropriate stats and mention in figure legends.

6) Few statements in the manuscripts both in the introduction and discussion require proper citations either for previous literature or refer figures of the manuscript or we show or our results suggest e.t.c; few examples are below: Page #2, line 49 - 51, Page #18, line 567-568, and Page #18, line 596-597 so on...

7) Please check for grammar including using appropriate determiner and passive voice choices. Also revise the manuscript for any typographical errors. Following are a few of those spelling errors;

Page #2, line 62: Our previous data

Page #2, line 79: was 5.2 years

Page #17, line 552: MBP

Author Response

1. We are aware of the limitations of our study and necessity to confirm our findings in vivo. However, the efficiency of neutralizing anti-CD49d (natalizumab) was already demonstrated in the clinical trials (Rommer 2014), similarly with the use of neutralizing mAbs anti-CD154 (clone IDEC-131) and anti-CD40 in human and EAE studies (Aarts 2017). Yet, no studies were done with blocking both CD49d and CD154. We showed in another manuscript that was in parallel sent to Cells (#632583, ‘Multiple Sclerosis CD49d+CD154+ As Myelin-Specific Lymphocytes Induced During Remyelination’) that maturing OPCs managed to induce CD49d+CD154+ lymphocyte proliferation in CD40-CD154-dependent manner. As this process takes place in CNS, the effectiveness of the therapy with the use of mAbs is limited. Therefore, in the next project we are planning to use anti-CD154 alone or in combination with anti-CD49d administrated intravenously and/or intraventriculary at different time points of EAE to estimate their influence on the disease inductions as well as remyelination process. The adequate paragraph describing the study limitation and future perspectives has been added to the Discussion section.

Our findings explain the phenomenon of insufficient remyelination based on the positive feedback between OPC-CD49d+CD154+ myelin-specific lymphocytes. Therefore, our data provide the opportunity for therapeutic interventions with using combined therapy with neutralizing mAbs anti-CD154 and CD49d. As CD49d+CD154+ lymphocyte proliferation might be induced by OPCs in CNS, the effectiveness of this therapy nowadays is limited. Further studies with the use of anti-CD154 mAbs alone as well as in combination with anti-CD49d mAbs administrated intravenously and/or intraventricularly at different time points of EAE to estimate their influence on the disease induction and remyelination process may indicate a new trend for therapeutic approach’.

 a. Rommer, P.S.; Dudesek, A.; Stüve, O.; Zettl, U.K. Monoclonal antibodies in treatment of multiple sclerosis. Clin Exp Immunol. 2014;175(3):373-384

b. Aarts, S.A.B.M.; Seijkens, T.T.P.; van Dorst, K.J.F.; Dijkstra, C.D.; Kooij, G.; Lutgens, E. The CD40–CD40L Dyad in Experimental Autoimmune Encephalomyelitis and Multiple Sclerosis. Front Immunol. 2017;12(8):1791

 2. We used a well-described protocol of EAE induction (Mendel 1995) employed in our laboratory for many years. EAE may be induced also in male mice, but for better results female mice weighing 19–20 g are preferred (Contarini 2017). A study by Dias and colleagues found that specifically female C57BL/6 mice immunized with MOG peptides presented typical MS pathology. Immunization of female C57BL/6 mice results in increased inflammatory cells infiltrate and increased cytokine levels in the CNS (Dias 2015) that was confirmed by us in immunohistochemistry. Moreover, using this model we confirmed the presence of CD154+CD49d+ cells that were colocalized within remyelinating plaque with maturating OPC (manuscript sent to Cells #632583, ‘Multiple Sclerosis CD49d+CD154+ As Myelin-Specific Lymphocytes Induced During Remyelination’). The appropriate citations have been added to the Materials and Methods.

Mendel, I.; Kerlero de Rosbo, N.; Ben-Nun, A. A myelin oligodendrocyte glycoprotein peptide induces typical chronic experimental autoimmune encephalomyelitis in H-2b mice: fine specificity and T cell receptor V beta expression of encephalitogenic T cells. Eur J Immunol. 1995;25(7):1951-1959. Contarini, G.; Giusti, P.; Skaper, S.D. Active Induction of Experimental Autoimmune Encephalomyelitis in C57BL/6 Mice. Methods Mol Biol. 2018;1727:353-360. Dias, A.T.; De Castro, S.B.; Alves, C.C.; Mesquita, F.P.; De Figueiredo, N.S.; Evangelista, M.G.; Castañon, M.C.; Juliano, M.A.; Ferreira, A.P. Different MOG(35-55) concentrations induce distinguishable inflammation through early regulatory response by IL-10 and TGF-β in mice CNS despite unchanged clinical course. Cell Immunol. 2015; 293(2):87-94.

3. We analyzed the influence of human CD49d+CD154+ lymphocytes and mouse brain CD3+ memory CD19+ cells on OPC viability/apoptosis/necrosis. Three independent methods were used: DIC macroscopy visualization (morphology), ANXV vs. PI fluorescence (apoptosis) and LDH assay (cell lysis). The data from these experiments are presented in the supplementary figure (S Figure 1A and B), but described in the main text (Results, page 8, lines 353-359 and 374-380). Additional information has been added to the Material and Method section  (page 4-5, lines 182-197).

2.10. OPC viability and apoptosis

The influence of mice and human lymphocytes on OPC viability was assessed using three independent methods which allowed to analyze OPC morphology, apoptosis/necrosis, and cell lysis.

2.10.1. DIC microscopy

The visualization of live cell interactions between lymphocytes and OPCs was performed on 8-well glass chamber slides (Nalge Nunc International, Waltham, MA, US). During the course of 21 hours, lymphocyte/OPC interactions were imaged at five time points (0, 1, 3, 5 and 7 hour intervals) with a Zeiss Axiovert 200 inverse microscope with a Zeiss LD Plan-Neofluar 40x/0.62 Ph2 Korr differential interference contrast objective (Göttingen. Germany).

2.10.2. OPC apoptosis

The binding of ANXV-FITC to phosphatidylserine was used as a sensitive measurement of OPC apoptosis. Additional staining with (PI) enabled to distinguish between early and late stage of apoptosis, as ANXV binds to both types of cells. After incubation of OPCs with lymphocytes, samples (100 mL) were washed twice in ice cold PBS without Ca+2/Mg+2 and incubated with ANXV-FITC and PI (BD Pharmingen) according to the manufacturer’s instructions. OPCs were identified and gated on the SSC/FSC dot plot and analyzed by flow cytometry.

2.10.3. Lactate dehydrogenase (LDH) release assay

LDH release was measured by colorimetric method using Cytotoxicity Detection Kit (Sigma-Aldrich, St. Louis, USA) according to the manufacturer’s instruction. Concentration of released LDH from 100% of cell lysis (OLs lysed with 1%Triton X-100) was used as the positive control and supernatants from OLs supplemented with 10% PBS as negative control (spontaneous LDH release). The rate of lysed cells was calculated based on the normalization of each sample to the level of LDH released by positive control subtracted from negative control samples. All samples were analyzed in duplicate.

4. We have improved the quality of Fig.1C and 2B as it was suggested. GFAP, as the sensitive marker of astrocyte cell line, and nestin, as the marker of early differentiation stage of OPCs, were used to confirm that glia are polarized into oligodendrocytes but not to astrocytes as well as to assess the effect of lymphocytes on this process. Therefore, in all settings GFAP signal should be negative, while nestin signal should be positive only in the samples of OPCs before PMA stimulations (marked in the Fig. 2B as “glia polarized into OPC” and in the Fig. 1C as “MO3.13 not polarized”). We also added statistical analysis of florescence intensity to demonstrate the differences in the Fig.1C and 2B according to the Reviewer’s suggestion.  Adequate statements describing the algorithm of florescence signal calculation was added to the Material and Method section.

Fluorescence intensity was determined as the average fluorescence (Avg. area), the sum of the fluorescence from all segments divided by the number of segments. The average fluorescence was calculated using at least twenty single cells taken from four independent experiments. The level of baseline fluorescence was established individually for each experiment. Nonspecific fluorescence (signal noise) was electronically diminished to the level when nonspecific signal was undetectable [9].’

5. In the Fig. 2, we did not show the control values of brain mononuclear cells as they do not infiltrate or occupy healthy brains. We previously visualized subpopulation of CD49d+CD154+ lymphocytes in the EAE brain in comparison to negative signal obtained from IHC of HC brain (manuscript sent to Cells #632583; Figure 3 in ‘Multiple Sclerosis CD49d+CD154+ As Myelin-Specific Lymphocytes Induced During Remyelination’).

6. Appropriate citations have been added:

Barnett, M.H.; Prineas, J.W. Relapsing and remitting multiple sclerosis: pathology of the newly forming lesion. Ann Neurol. 2004; 55:458–468. Ffrench-Constant, C.; Raff, M.C. Proliferating bipotential glial progenitor cells in adult rat optic nerve. Nature. 1986;319(6053):499-502. Dawson, M.R.; Polito, A.; Levine, J.M.; Reynolds, R. NG2-expressing glial progenitor cells: an abundant and widespread population of cycling cells in the adult rat CNS. Mol Cell Neurosci. 2003; 24(2):476-88. Miller, T.; Williams, K.; Johnstone, R.W.; Shilatifard, A. Identification, cloning, expression, and biochemical characterization of the testis-specific RNA polymerase II elongation factor ELL3. J Biol Chem. 2000; 275(41):32052-32056. Mendel, I.; Kerlero de Rosbo, N.; Ben-Nun, A. A myelin oligodendrocyte glycoprotein peptide induces typical chronic experimental autoimmune encephalomyelitis in H-2b mice: fine specificity and T cell receptor V beta expression of encephalitogenic T cells. Eur J Immunol. 1995; 25(7):1951-1959. Contarini, G.; Giusti, P.; Skaper, S.D. Active Induction of Experimental Autoimmune Encephalomyelitis in C57BL/6 Mice. Methods Mol Biol. 2018; 1727:353-360. Dias, A.T.; De Castro, S.B.; Alves, C.C.; Mesquita, F.P.; De Figueiredo, N.S.; Evangelista, M.G.; Castañon, M.C.; Juliano, M.A.; Ferreira, A.P. Different MOG(35-55) concentrations induce distinguishable inflammation through early regulatory response by IL-10 and TGF-β in mice CNS despite unchanged clinical course. Cell Immunol. 2015; 293(2):87-94.

As in this manuscript we have cited publication being still under review in Cells (Number 632583, ‘Multiple Sclerosis CD49d+CD154+ As Myelin-Specific Lymphocytes Induced During Remyelination’), we have highlighted these citations in the main text in yellow till the final decision.

We have checked and corrected language mistakes.

Round 2

Reviewer 1 Report

Excellent paper, as in its revised draft